# Use of seat belts among public transport drivers in Tacna, Peru: Prevalence and risk factors

**Armando Miñan-Tapia**[1]*, **Gloria S. Torres-Riveros**[1], **José Choque-Vargas**[1], **Madeleyni Aycachi-Incacoña**[2], **Neil Flores-Valdez**[2], **Orlando Vargas-Anahua**[1], **Christian R. Mejia**[3]

1 Escuela Profesional de Medicina Humana, Universidad Privada de Tacna, Tacna, Perú, 2 Universidad Nacional Jorge Basadre Grohmann, Tacna, Perú, 3 Translational Medicine Investigation Centre, Universidad Norbert Wiener, Lima, Perú

* arluminan@gmail.com

## Abstract

### Introduction

There are individuals who still refuse to wear seat belts, despite its effectiveness in reducing morbidity and mortality in road traffic accidents. We aimed to determine the prevalence and risk factors according to the use of seat belts among public transport drivers in Tacna, Peru.

### Methodology

This analytical transversal study was carried out among public transport drivers (buses and taxis) in a Peruvian city. Questionnaires were used to evaluate the general and occupational characteristics and the use of seat belts (observed). Descriptive statistics and risk factors were obtained, these latter through generalized linear models.

### Results

Of the 460 drivers, 77% used their seat belts, with a difference in use depending on the type of public transport (p<0.001). In the multivariate model, the risk of not using the belt was associated with the following: older age (p<0.001), having complete studies (p<0.001), a higher level/category of driving license (3 categories had p<0.001), having a higher number of previous road traffic accidents (p = 0.011), and received medical attention in that accident (p<0.001), those who reported using a cell phone while driving (p = 0.005), if the co-driver's belt had 3 anchorage points (p<0.001), and working for > 5 hours that day (p = 0.002). However, male drivers and those who had their belt with 3 anchorage points had greater use (both p<0.001).

### Conclusions

One in five drivers did not use a seat belt, and important characteristics of those who did not comply with this traffic law were evaluated to generate control and intervention measures.

**Data Availability Statement:** All relevant data are within the paper and its Supporting Information files.

**Funding:** The author(s) received no specific funding for this work.

**Competing interests:** The authors have declared that no competing interests exist.

## Introduction

Road traffic crashes are a major public health problem, representing one of the most frequent causes of traumatic injuries and deaths worldwide [1, 2]. The seat belt is a safety feature, which reduces the risk of serious injury [3, 4] and death from road traffic crashes [5, 6]. Its use is mandatory for all vehicle occupants due to the legislation in force, and this is applied in several countries worldwide [1]. The prevalence of seat belt use varies to worldwide, as 48% [7], 73% [8], 58% [9], 82% [10], and 86–90% [11–13] were reported in Bosnia-Herzegovina, Korea, Iran, Singapore, and in the United States of America, respectively. On the contrary, in Latin American countries, the prevalence of 82% was reported in Ecuador [14], 36% in Colombia [15] and 38% in Mexico [16]; lower prevalence was observed among co-drivers [15, 16]. However, in another Peruvian city, a prevalence of 79% was reported among public transport drivers [17].

Several factors have been found to increase seat belt use, such as being in a city far from the capital, being female, driving a taxi [15], older age [10], and being more educated [7, 9, 11]. However, most of those studies were conducted using secondary databases [8, 10, 12, 13], through observations from outside the vehicles [7, 15–17] or through self-reports by the drivers [9, 11, 12, 14]. Therefore, it is necessary to conduct field research that directly observes their use and the most important influences, especially in low- and middle-income countries such as Peru [1]. The objective of our study was to determine the prevalence and risk factors to the use of seat belts among public transport drivers in Tacna, Peru.

## Methodology

### Study design, location and time

An observational, analytical, and cross-sectional study was conducted in the city of Tacna, located in southern Peru, during the years 2018 and 2019.

### Population, sample and sampling

The study population consisted of public transport drivers (buses and taxis), who worked/resided in Tacna city. First, a pilot was carried out among 30 public transport drivers (not included in the study) in order to calculate the sample size, with statistical power and significance of 80% and 95%, respectively. A minimum sample size of 448 drivers was obtained (adding 10% for possible losses), and they were selected through a systematic random sampling at the main stops in the city.

Drivers who agreed to participate in the study and were 18 years or older were included, while those who were off duty (at the time of the survey) were excluded, as were the collection sheets with inconsistent data (exclusion was less than 5% of the entire sample).

### Instrument

The questionnaire was designed by the authors of this study (*ad hoc)* (S1 File), and a pilot was pre-test to assess the relevance and understanding of the questions in 30 drivers who did not participate in the study. The answers provided by the drivers were obtained to confirm that the questions were understandable.

The questionnaire included the following items: characteristics given at the time of the survey, general and work characteristics of the drivers, previous traffic accident reports, mobile phone use, and seat belt use.

The general characteristics were evaluated, such as sex, age, and degree of education (re-categorized as incomplete/complete). For the labor characteristics, we inquired about the type of

transport service (taxi/bus), the characteristics of their work, category of driving license they had, the number of working hours at the time of the survey (re-categorized as less or equal than 5 hours and more than 5 hours, which is the legal maximum for continuous driving hours), the number of daily work hours on average, and years of working experience as drivers. In addition, we asked the drivers if they had been in a traffic accident and whether they had received medical care because of that accident. We also asked whether they had ever used a mobile phone while driving and observed its use during the interview (this is a subject of another study).

For evaluation to seat belt use (dependent variable), we observed if the driver used seat belts (no/yes), if not use, the reasons for not using it were asked, and they were recommended to do so. We checked if the co-driver's seat was equipped with a seat belt (no/yes). For both safety belts (driver and co-driver), the types of anchorage they had were observed (2 or 3 anchorage points).

## Procedures

Data was collected through the field work carried out by the research group and other collaborators, previously trained (denominated interviewers), each of whom objectively observed and obtained answers that were noted in the questionnaire. Each pollster selected the public transport vehicle to be evaluated using a systematic random sampling method, by getting into the first vehicle that transited in service at the authorized stops for picking up passengers (buses) and the second vehicle for the taxi drivers, considering that any vehicle could be selected. The interviewer got into the vehicle and moved close to the front of the vehicles to make a first observation of the driver's seat belt use. When the vehicle was parked, at a traffic light, or when it stopped, the interviewer explained the objectives of the study to the driver and asked for verbal consent (to avoid the information bias of the dependent variables).

The data was then entered into a data sheet in the Excel 2013 program. The questionnaires were numbered during the data collection (consecutively), this allowed the lead author perform a quality control of the data.

## Statistical analysis

After data quality control (using Excel 2013), the data was then entered into the Stata v11.0 statistical program. Descriptive statistics of the drivers' general characteristics were analyzed, the frequencies and percentages for the categorical variables were evaluated, and after assessing the normality of the continuous variables, the most appropriate measures of central tendency and dispersion were described. Afterwards, the characteristics of the drivers were compared according to their type of public transport, and the p value was calculated using the Chi-square and Student T tests, according to the type of variable.

The dependent variable was the seat belt use that were observed during the interviewer's first evaluation. Statistical analysis was carried out using generalized linear models (Poisson family models, the log-link function, and robust models, using the type of public transport as a cluster) [18–21], evaluating the raw and adjusted relative risks, the 95% confidence intervals, and the p-value of each crossing. We considered p values <0.05 as statistically significant.

## Ethics

The ethics committee of the "Hospital Nacional Docente Madre Niño" in Peru evaluated and approved this project. The authors of the study got a verbal consent from the participants. This verbal consent was obtained before the application of the study. It was witnessed by the pollsters, but it was not documented (to reduce rejection rate). The significance of this study was

explained to the drivers, then verbal informed consent was requested (after evaluating the main variables), and the obtained information was anonymized by assigning a unique number to each driver.

## Results

Of the 460 public transport drivers assessed, 440 (95.7%) were males, with an average age of 41.9 (standard deviation: ± 9.7 years). About the type of public transport evaluated, 227 of them drove taxis (49.4%) and 233 drove buses (50.6%). Median working hours at the time of evaluation was 5 hours (range: 4–8 hours), and they reported working a median of 12 hours daily (range: 10–14 hours), with 9 years working experience as drivers (range: 5–15 years). Of the total drivers assessed, 77% (356/460) used seat belts, of which 44.4% were taxi drivers and 55.6% were bus drivers (cf **Table 1**).

In the Table 2 we can see the main characteristics of the drivers according to the type of public transport in Tacna-Peru, as well as the main differences between these groups.

When the drivers were asked why they did not use their seat belts (104/460 drivers), 46.2% said it was because of perceived discomfort and 30.8% said because of lack of habit (**Fig 1**).

The bivariate analysis identified some factors associated with seat belt use in public transport drivers. Similarly, it was identified that risk factors decreased with the use of seat belts (**Table 3**).

The multivariate analysis identified male drivers and having seat belts with three anchorage points as factors that increased the use of seat belts. On the contrary, older age, having complete studies, higher classification of driving license, greater number of road traffic accidents as well as having received medical attention because of the accidents, those who reported using the cell phone while driving, vehicles where the co-driver's belt had 3 anchorage points, and having worked more than 5 hours at the time of the evaluation, were identified as risk factors that decreased the use of the seat belt in public transport drivers (cf. **Table 4**).

## Discussion

### Main findings

We found that 4 out of 5 drivers evaluated used seat belts (77.4%), with differences depending on the type of public transport evaluated. Furthermore, 7 out of 10 taxi drivers used their belts, compared to bus drivers, where 9 out of 10 used the belt. We identified risk factors that modified the frequency of seat belt use in public transport drivers in one Peruvian city.

### Prevalence to seat belt use

It was shown that 4 out of 5 public transport drivers were using seat belts during their workday, differing according to the type of public transport evaluated (buses: 56% vs. taxis: 44%; $p < 0.001$). In our population, the average age and number of road traffic accidents of bus drivers was higher than that of taxi drivers ($p < 0.001$). We found that 23% of drivers did not use seat belts, and this finding is similar to that of a study of another Peruvian city, where 21% of public transport drivers did not use seat belts [17]. Contrarily, the values are lower than those observed in Bosnia-Herzegovina, where 52% of drivers did not use seat belts in urban areas [7], as well as 30% reported in an African country [22], 62% in two cities in Mexico [16], 63% in two cities in Colombia (with lower use of vans and cars compared to taxis) [15]. Furthermore, in Ethiopia, taxi drivers had twice as much seat belt use as minibus drivers ($p = 0.004$) [22]. These discrepancies may be due to the different ways and areas in which seat belt use was assessed, as well as cultural, legal and even population differences, and the

**Table 1. General characteristics of the drivers evaluated.**

| Variable | N | % |
|---|---|---|
| **Sex** | | |
| Female | 20 | 4.4 |
| Male | 440 | 95.6 |
| **Age (years)¶** | 41.9 | ± 9.7 |
| **Studies** | | |
| Incompleted | 131 | 28.5 |
| Completed | 329 | 71.5 |
| **Driving license (level)** | | |
| A I | 1 | 0.22 |
| A II a | 110 | 23.9 |
| A II b | 183 | 39.8 |
| A III a | 54 | 11.7 |
| A III b | 14 | 3.1 |
| A III c | 98 | 21.3 |
| **Previous traffic fine** | | |
| No | 189 | 41.1 |
| Yes | 271 | 58.9 |
| **Number of previous road accidents$^{\beta}$** | 1 | 0–2 |
| **Medical attention for road accidents** | | |
| No | 185 | 68.3 |
| Yes | 86 | 31.7 |
| **Type of anchorage of seat belt (drivers)** | | |
| 2 points | 69 | 15 |
| 3 points | 391 | 85 |
| **Type of anchorage of seat belt (co-drivers)** | | |
| 2 points | 115 | 25 |
| 3 points | 345 | 75 |
| **Hours of work at the time of evaluation** | | |
| Less or equal than 5 hours | 243 | 52.8 |
| More than 5 hours | 217 | 47.2 |
| **Seat belt use** | | |
| No | 104 | 22.6 |
| Yes | 356 | 77.4 |
| **Type of public transport** | | |
| Taxis | 227 | 49.4 |
| Buses | 233 | 50.6 |

¶ Mean and standard deviation.

$^{\beta}$ Median and interquartile range.

possibility of overconfidence. Therefore, it is important to make a situational analysis of each reality in order to promote massive strategies to increase the prevalence of seat belt use in the future, which has been shown to decrease mortality from road traffic accidents [23].

When asked why drivers did not wear their seat belts, they said it was because of the perceived discomfort in front of the belt. These findings are consistent with those reported in studies conducted in Iran [24] and Peru [17]. In other settings, drivers who did not use their seatbelts cited disbelief in the safety of the device as a reason for not using it; therefore, they

**Table 2. Characteristics of the drivers according to the type of public transport in Tacna-Peru.**

| Variable | Type of public transport (drivers) | | p value* |
|---|---|---|---|
| | Taxis | Buses | |
| **Sex** | | | |
| Female | 19 (95.0) | 1 (5.0) | < 0.001 |
| Male | 208 (47.3) | 232 (52.7) | |
| **Age (years) ⁋** | 38.8 (0.6) | 44.8 (0.6) | < 0.001** |
| **Studies** | | | |
| Incompleted | 23 (17.6) | 108 (82.4) | < 0.001 |
| Completed | 204 (62.1) | 125 (37.9) | |
| **Driving license (level)** | | | |
| A I | 1 (100.0) | 0 (0.0) | < 0.001 |
| A II a | 110 (100.0) | 0 (0.0) | |
| A II b | 78 (42.6) | 105 (57.4) | |
| A III a | 3 (5.6) | 51 (94.4) | |
| A III b | 2 (14.3) | 12 /85.7) | |
| A III c | 33 (33.7) | 65 (66.3) | |
| **Previous traffic fine** | | | |
| No | 93 (50.8) | 90 (49.2) | 0.608 |
| Yes | 134 (48.4) | 143 (51.6) | |
| **Number of previous road accidents⁋** | 0.8 (0.1) | 1.6 (0.1) | < 0.001** |
| **Medical attention for road accidents** | | | |
| No | 60 (32.4) | 125 (67.6) | 0.131 |
| Yes | 36 (41.9) | 50 (58.1) | |
| **Reported having used a cell phone while driving (ever)** | | | |
| No | 83 (40.7) | 121 (59.3) | 0.001 |
| Yes | 144 (56.3) | 112 (43.7) | |
| **Type of anchorage of seat belt (drivers)** | | | |
| 2 points | 12 (17.4) | 57 (82.6) | < 0.001 |
| 3 points | 215 (54.9) | 176 (45.1) | |
| **Type of anchorage of seat belt (co-drivers)** | | | |
| 2 points | 12 (10.4) | 103 (89.6) | < 0.001 |
| 3 points | 215 (62.3) | 130 (37.7) | |
| **Hours of work at the time of evaluation** | | | |
| Less or equal than 5 hours | 119 (48.9) | 124 (51.1) | 0.864 |
| More than 5 hours | 108 (49.8) | 109 (50.2) | |
| **Seat belt use** | | | |
| No | 69 (66.4) | 35 (33.6) | < 0.001 |
| Yes | 158 (44.4) | 198 (55.6) | |

⁋ Mean and standard deviation; p values obtained by *Chi square / ** Student´s T.

consider it a waste of time [22]. In a study in Colombia, drivers reported simulating the use of their seat belts, using it when they see a traffic policeman or when driving through areas where they could be found, stating that it causes them discomfort [15], and this could be related to the state of the seat belt, and this issue could be reduced by using comfort devices. Similarly, in our study, we observed that drivers pretended to use their belts; therefore, unannounced checks from inside the vehicles (by law enforcers) could increase their use and compliance with the legislation [25].

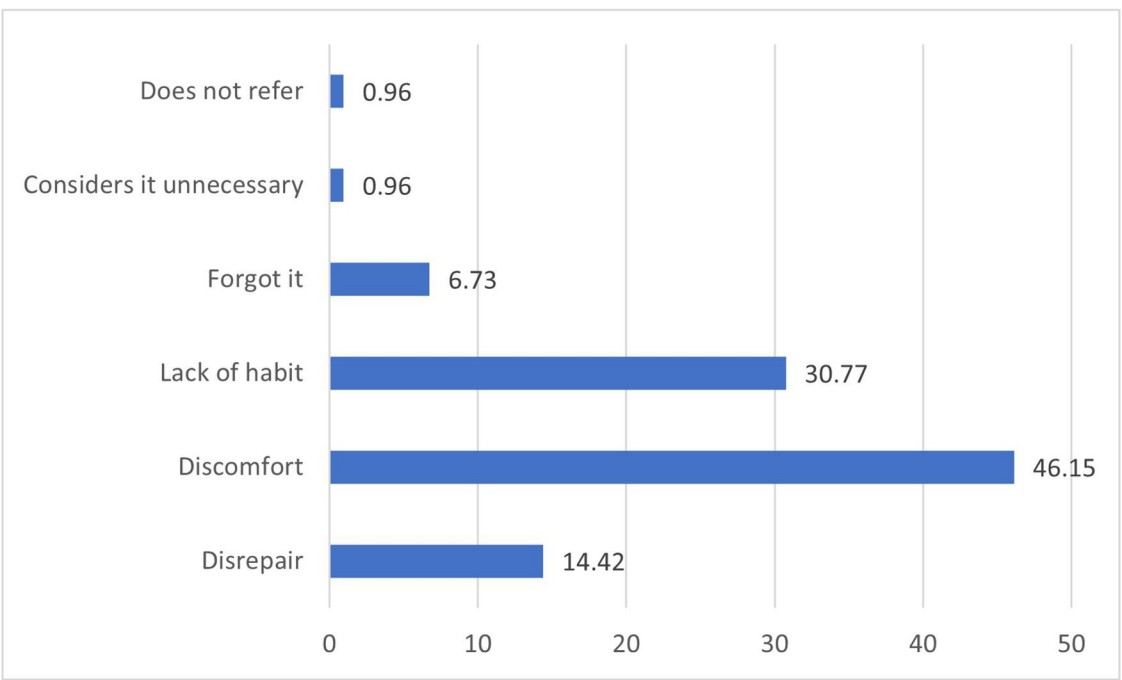

**Fig 1. Reasons given by drivers for why they were not wearing their seat belts.**

### Risk factors associated to seat belt use

Older people were identified as having less seat belt use, and this contradicts a study by Greene and Smith, who found that a geriatric population over the age of 75 had a higher rate of seat belt use compared to younger people [26]. Similarly, a study in Colombia reported that people over 60, although they drove less, made the most use of safety devices [15]. Thus, developing interventions (such as seat belt reminders in urban settings) and promoting road safety education programs at early ages may yield greater long-term benefits by promoting greater seat belt use [27, 28]. This is contrary to some accredited driver education interventions, who refer do so in order to reduce the number or cost of their traffic fine [25]. It is important to promote a good road safety culture in the general population, which could be reflected over time.

We also found that drivers who reported having completed studies and/or having a higher level/category of driving license used seat belts less, and this is contrary to what was found in Ethiopia, where a higher level of education was positively associated with seat belt use [22]. This association was also reported in the United States [11] and in Bosnia-Herzegovina [7], where higher educational level was a protective factor for increased seat belt use in both urban and rural areas. This may be due to the fact that more educated drivers feel more experienced and less likely to be victims of road traffic accidents, being able to lead overconfidence, thus underestimating the usefulness of seat belts, which may also be due to fatigue over the years. These unexpected results will have to be assessed later in order to target future interventions.

Having more road traffic injuries, with some of them requiring medical attention, was associated with less seat belt use. Seat belt use effectively prevents fatality and serious injury [8, 29]; however, it was noted in our study that despite having more traumatic histories, drivers were not using seat belts. This is similar to what Zabihi et al. reported, where they found a low rate of seat belt use despite the fact that 36% of their population had a history of road traffic accidents [24]. This finding suggested that, despite having a traumatic experience (such as road

**Table 3. Bivariate analysis of the risk factors associated with the use of seat belts among public transport drivers in Tacna-Peru.**

| Variable | Seat belt use | | Bivariate analysis | | |
|---|---|---|---|---|---|
| | Yes | No | cRR | CI95% | p value |
| **Sex** | | | | | |
| Female | 7 (35.0) | 13 (65.0) | Comparison category | | |
| Male | 349 (79.3) | 91 (20.1) | 2.27 | 1.59–3.23 | < 0.001 |
| **Age (years)** | 42.3 | 40.4 | 1.00 | 1.00–1.01 | < 0.001 |
| **Studies** | | | | | |
| Incompleted | 109 (83.2) | 22 (16.8) | Comparison category | | |
| Completed | 247 (75.1) | 82 (24.9) | 0.90 | 0.82–0.99 | 0.027 |
| **Driving license (level)** | | | | | |
| A I | 1 (100.0) | 0 (0.0) | Comparison category | | |
| A II a | 59 (53.6) | 51 (46.4) | The result does not converge | | |
| A II b | 163 (89.1) | 20 (10.9) | 0.89 | 0.79–0.99 | 0.035 |
| A III a | 47 (87.0) | 7 (13.0) | 0.87 | 0.83–0.92 | < 0.001 |
| A III b | 11 (78.6) | 3 (21.4) | 0.79 | 0.64–0.96 | 0.020 |
| A III c | 75 (76.5) | 23 (23.5) | 0.77 | 0.59–0.98 | 0.035 |
| **Previous traffic fine** | | | | | |
| No | 136 (74.3) | 47 (25.7) | Comparison category | | |
| Yes | 220 (79.4) | 57 (20.6) | 1.07 | 1.04–1.10 | < 0.001 |
| **Number of previous road accidents** | 1.2 (0.08) | 1.0 (0.13) | 1.02 | 0.94–1.11 | 0.610 |
| **Medical attention for road accidents** | | | | | |
| No | 163 (88.1) | 22 (11.9) | Comparison category | | |
| Yes | 60 (69.8) | 26 (30.2) | 0.79 | 0.61–1.03 | 0.083 |
| **Reported having used a cell phone while driving (ever)** | | | | | |
| No | 180 (88.2) | 24 (11.8) | Comparison category | | |
| Yes | 176 (68.8) | 80 (31.2) | 0.78 | 0.63–0.97 | 0.025 |
| **Type of anchorage of seat belt (drivers)** | | | | | |
| 2 points | 47 (68.1) | 22 (31.9) | Comparison category | | |
| 3 points | 309 (79.1) | 82 (20.9) | 1.16 | 1.01–1.34 | 0.040 |
| **Type of anchorage of seat belt (co-drivers)** | | | | | |
| 2 points | 96 (83.5) | 19 (16.5) | Comparison category | | |
| 3 points | 260 (75.4) | 85 (24.6) | 0.90 | 0.86–0.94 | < 0.001 |
| **Hours of work at the time of evaluation** | | | | | |
| Less or equal than 5 hours | 188 (77.4) | 55 (22.6) | Comparison category | | |
| More than 5 hours | 168 (77.4) | 49 (22.6) | 1.00 | 0.87–1.16 | 0.993 |
| **Drivers** | | | | | |
| Taxis | 158 (69.6) | 69 (30.4) | Comparison category | | |
| Buses | 198 (84.9) | 35 (15.1) | 1.22 | 1.22–1.22 | < 0.001 |

Crude relative risks, 95% confidence intervals and p-values were obtained with the Poisson family models, the log-link function, and robust models, using the type of public transport (bus or taxi) as a cluster; all with the generalized linear models.

traffic accidents), some drivers are still unaware of the importance and usefulness of the seat belt.

Those who reported using a cell phone while driving had 20% less seat belt use, similar results were reported in an observational study in Doha, where one in five drivers who used a phone did not use a seat belt, with a higher proportion of the population as seat belt users (p < 0.001) [30]. This was also reported in Florence, where drivers who used phones while

**Table 4. Multivariate analysis of risk factors associated with seat belt use among public transport drivers in Tacna-Peru.**

| Variables | Relative Risk | Confidence Interval 95% | p value |
|---|---|---|---|
| **Male sex** | 2.19 | 2.09–2.30 | <0.001 |
| **Age** (years) | 0.997 | 0.997–0.997 | <0.001 |
| **Completed studies** | 0.99 | 0.98–0.99 | <0.001 |
| **Driving license (level)** | | | |
| A I | Comparison category | | |
| A II a | 0.72 | 0.71–0.72 | <0.001 |
| A II b | 0.81 | 0.78–0.84 | <0.001 |
| A III a | 0.78 | 0.75–0.81 | <0.001 |
| A III b | 0.95 | 0.76–1.17 | 0.612 |
| A III c | 0.77 | 0.60–0.99 | 0.043 |
| **Have a previous traffic fine** | 0.99 | 0.89–1.11 | 0.882 |
| **Number of previous road accidents** | 0.97 | 0.95–0.99 | 0.011 |
| **Medical attention for road accidents** | 0.86 | 0.81–0.93 | <0.001 |
| **Reported having used a cell phone while driving** | 0.80 | 0.68–0.93 | 0.005 |
| **Driver's belt with 2 anchorage points** | 1.50 | 1.31–1.73 | <0.001 |
| **Co-driver's belt with 3 anchorage points** | 0.77 | 0.77–0.77 | <0.001 |
| **More than 5 hours of work at the time of evaluation** | 0.97 | 0.96–0.99 | 0.002 |

Relative risks, 95% confidence intervals and p-values were obtained with the Poisson family models, the log-link function, and robust models, using the type of public transport (bus or taxi) as a cluster; all with the generalized linear models.

driving were 12% less likely to wear seat belts [31]. Similarly, in Saudi Arabia, drivers who had mobile phone-related crashes had less seat belt use [32]. These studies show that drivers who use or report using a mobile phone while driving are less likely to wear a seat belt, although the studies were conducted in different countries, these findings are similar with an observing further that, in some cities where phone use while driving was banned, there was a reduction in driver fatalities [33].

In our study, regarding the type of anchorage, the safety belts evaluated had mainly 3 points, even though some vehicles still had 2 anchorage points (exceptionally allowed for vehicles manufactured before 1980, according to the road safety regulations) [34]. It was observed that drivers who had triple anchorage points on their belts used them more; however, if the co-driver had three anchorage points, they used them less. We found no study to compare this finding with; however, a study in Australia reported that seat belt misuse by both adults and children is very common, regardless of the anchorage points [35]. Likewise, in Ecuador, two investigations conducted to evaluate the resistance of safety belts use reported that the manufacture of these safety devices was not in accordance with the minimum standards of road safety, being a country where two and three anchorage points are mostly used [36, 37]. Although our study did not investigate the characteristics of the belt or the manufacture year of the vehicles, we observed that the belts in some vehicles were modified or adapted, this finding could be decreasing intrinsic properties of the seat belts and its effect in the decrease of mortality, and it could be a reason for future investigations.

In our study, we found that those taxi drivers who had worked for more than 5 hours (on the day of the evaluation) used their seat belt less, although this association has not been reported in previous studies. A study carried out in Argentina reported that 60% of taxi drivers worked more than 10 hours per day, and in that same population, 20% never use seat belts [38]. A longer working time can expose them to a greater work overload, which would be reflected in a lower use of the safety belt and with the potential appearance of health problems

like the Burnout Syndrome in drivers of public transportation, evidenced by a study in another Peruvian city, where 20% of drivers had more than 10 years of service [39]. Both findings are risk factors for road traffic accidents [40], which should be evaluated and controlled early.

## Limitations and strengths

We had as limitations, the time available to public transport drivers to participate in the evaluation, which inevitably reduced the population size; however, the minimum sample size was reached through a probability sampling, which allows us to see the reality of public transport drivers in the city of Tacna. To reduce the observation bias, the use of seat belts was assessed from inside the vehicle while going up to the transport unit. In the case of drivers who did not use a seatbelt, the evaluators asked why they did not use this device and recommended using it at the beginning of the interview.

## Conclusions

Four out of five public transport drivers in Tacna used seat belts, with differences depending on the type of public transport. Similarly, factors that modified the frequency of seat belt use in public transport drivers in one Peruvian city were identified. These results show the reality of safety belts use in public transport drivers, allowing the authorities to apply control measures to increase the use. These measures can reduce the morbidity and mortality burden of road traffic accidents, which represent a major problem in our country.

## Supporting information

**S1 File. Questionnaire.**
(PDF)

**S2 File.**
(PDF)

## Acknowledgments

We acknowledgment to the collaborators Dariela Vizcarra Jiménez, Shadya Oviedo Yui, Yasmin Valencia and Sué Torres López, who actively participated in the project phase of research and data collection.

## Author Contributions

**Conceptualization:** Armando Miñan-Tapia, Gloria S. Torres-Riveros, Orlando Vargas-Anahua.

**Data curation:** Armando Miñan-Tapia, Gloria S. Torres-Riveros, José Choque-Vargas, Madeleyni Aycachi-Incacoña.

**Formal analysis:** Neil Flores-Valdez, Christian R. Mejia.

**Investigation:** Armando Miñan-Tapia, Gloria S. Torres-Riveros, José Choque-Vargas, Madeleyni Aycachi-Incacoña, Neil Flores-Valdez, Orlando Vargas-Anahua, Christian R. Mejia.

**Methodology:** Armando Miñan-Tapia, Gloria S. Torres-Riveros, José Choque-Vargas, Madeleyni Aycachi-Incacoña, Neil Flores-Valdez, Orlando Vargas-Anahua, Christian R. Mejia.

**Writing – original draft:** Armando Miñan-Tapia, Gloria S. Torres-Riveros, José Choque-Vargas, Madeleyni Aycachi-Incacoña, Neil Flores-Valdez, Orlando Vargas-Anahua, Christian R. Mejia.

**Writing – review & editing:** Armando Miñan-Tapia, Gloria S. Torres-Riveros, José Choque-Vargas, Madeleyni Aycachi-Incacoña, Neil Flores-Valdez, Orlando Vargas-Anahua, Christian R. Mejia.

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
