## [Decision Letter · Decision Letter 0]

26 Oct 2020

PONE-D-20-30480

Use of seat belts among public transport drivers in Tacna, Peru: Prevalence and risk factors

PLOS ONE

Dear Dr. Miñan-Tapia,

Thank you for submitting your manuscript to PLOS ONE. After careful consideration, we feel that it has merit but does not fully meet PLOS ONE’s publication criteria as it currently stands. Therefore, we invite you to submit a revised version of the manuscript that addresses the points raised during the review process.

We look forward to receiving your revised manuscript.

Kind regards,

Feng Chen

Academic Editor

PLOS ONE

Journal Requirements:

2. In the ethics statement in the Methods section and online submission information, please specify the type of informed consent that was obtained from the participants (for instance, written or verbal, and if verbal, how it was documented and witnessed).

3. Please include additional information regarding the survey or questionnaire used in the study and ensure that you have provided sufficient details that others could replicate the analyses. For instance, if you developed a questionnaire as part of this study and it is not under a copyright more restrictive than CC-BY, please include a copy, in both the original language and English, as Supporting Information, or include a citation if it has been published previously.

4. In the Methods, please discuss whether and how the questionnaire was validated and/or pre-tested. If these did not occur, please provide the rationale for not doing so.

5.  We suggest you thoroughly copyedit your manuscript for language usage, spelling, and grammar. If you do not know anyone who can help you do this, you may wish to consider employing a professional scientific editing service.  

Reviewers' comments:

Reviewer's Responses to Questions

**Comments to the Author**

1. Is the manuscript technically sound, and do the data support the conclusions?

Reviewer #1: Partly

Reviewer #2: Partly

2. Has the statistical analysis been performed appropriately and rigorously? 

Reviewer #1: N/A

Reviewer #2: Yes

3. Have the authors made all data underlying the findings in their manuscript fully available?

Reviewer #1: Yes

Reviewer #2: Yes

4. Is the manuscript presented in an intelligible fashion and written in standard English?

Reviewer #1: No

Reviewer #2: Yes

5. Review Comments to the Author

Reviewer #1: This paper investigates the prevalence and influencing factors of seat-belts usage among public transport drivers in Tacna, Peru. The research topic is interesting and worth of investigation. However, the introduction on the proposed generalized linear model is somewhat vague. The authors are suggested to explicitly specify the formulations of the proposed methods and illustrate why they are suitable to model the use of seat belts. Moreover, some references on univariate/bivariate/multivariate generalized linear models should be added, such as:

A multivariate random parameters Tobit model for analyzing highway crash rate by injury severity. Accident Analysis and Prevention, 2017, 99: 184-191.

Jointly modeling area-level crash rates by severity: A Bayesian multivariate random-parameters spatio-temporal Tobit regression. Transportmetrica A: Transport Science, 2019, 15(2): 1867-1884.

Spatial joint analysis for zonal daytime and nighttime crash frequencies using a Bayesian bivariate conditional autoregressive model. Journal of Transportation Safety and Security, 2020, 12(4): 566-585.

Besides, the English writing should be improved.

Reviewer #2: The topic of this paper is interesting. The methods sound. The results are meaningful and useful. There are several suggestions to improve this paper.

1. More discussion about the methods used in this field is needed. And more references about different methods could be added. For example, the following ones about logit models are needed.

[1] Investigating the Impacts of Real-Time Weather Conditions on Freeway Crash Severity: A Bayesian Spatial Analysis, International Journal of Environmental Research and Public Health, 2020, 17(8): 2768.

[2] Investigation on the Injury Severity of Drivers in Rear-End Collisions Between Cars Using a Random Parameters Bivariate Ordered Probit Model, International Journal of Environmental Research and Public Health, 2019, 16(14) , 2632.

2. A table of the basic statistical information of the variables are needed.

3. The English writting could be improved.

6. PLOS authors have the option to publish the peer review history of their article (what does this mean?). If published, this will include your full peer review and any attached files.

Reviewer #1: No

Reviewer #2: No

---

## [Author Response · Author response to Decision Letter 0]

2 Apr 2021

1. Comments of the reviewers: In the ethics statement in the Methods section and online submission information, please specify the type of informed consent that was obtained from the participants (for instance, written or verbal, and if verbal, how it was documented and witnessed).

Comments of the authors: The authors of the study got a verbal consent from the participants. This verbal consent was obtained before the application of the study. It was witnessed by the pollsters, but it was not documented (to reduce rejection rate). Outcome evaluation (use the cellphone and seat belt use) was performed since the pollsters got into the vehicle (to avoid information bias), being one of the first reports of the use of seat belt in public transport drivers assessed from inside the vehicle 

2. Comments of the reviewers: Please include additional information regarding the survey or questionnaire used in the study and ensure that you have provided sufficient details that others could replicate the analyses. For instance, if you developed a questionnaire as part of this study and it is not under a copyright more restrictive than CC-BY, please include a copy, in both the original language and English, as Supporting Information, or include a citation if it has been published previously.

Comments of the authors: We have provided more information about the scale and we will be sending the scale in the support information, in its original version (Spanish) and in the English language.

3. Comments of the reviewers: In the Methods, please discuss whether and how the questionnaire was validated and/or pre-tested. If these did not occur, please provide the rationale for not doing so.

Comments of the authors: We have clarified this item in the methods section.

4. Comments of the reviewers: We suggest you thoroughly copyedit your manuscript for language usage, spelling, and grammar. If you do not know anyone who can help you do this, you may wish to consider employing a professional scientific editing service. 

Comments of the authors: We have improved it. 

5. Comments of the reviewers: the introduction on the proposed generalized linear model is somewhat vague.The authors are suggested to explicitly specify the formulations of the proposed methods and illustrate why they are suitable to model the use of seat belts.

Comments of the authors: The generalized linear regression model was performed, using the Poisson family with robust variance to find the risk factors associated with greater or lesser use of a seat belt (outcome / main / dependent variable). This model was preferred due to the study design (cross-sectional), due to the dichotomous outcome, and to unify the statistical models, relating the random variable (use of a seat belt) with the systematic component (other independent variables evaluated), thus adjusting the model by potential confounding variables. This generalized linear model is proposed by the Peruvian transport authorities to make estimates of accidents and other issues related to road safety in transport, [1] as well as studies that evaluate behaviors dependent on drivers. [2]

Our study does not directly assess the characteristics of traffic accidents, nor the trend of these, however, the use of seat belts is part of the road safety devices that can have a decrease in the rates of morbidity and mortality in the drivers.

6. Comments of the reviewers: Some references on univariate/bivariate/multivariate generalized linear models should be added.

Comments of the authors: We have added the suggested references. 

7. Comments of the reviewers: More discussion about the methods used in this field is needed. And more references about different methods could be added.

Comments of the authors: We have added the suggested references.

8. Comments of the reviewers: A table of the basic statistical information of the variables are needed.

Comments of the authors: We have added the basic statistical information of the variables evaluated in public transport drivers in table 01

9. Comments of the reviewers: The English writting could be improved.

Comments of the authors: We have improved it.

References

1. Ministerio de Transportes y Comunicaciones. Manual de Seguridad Vial. Lima - Perú; 2017. Avaible in: https://portal.mtc.gob.pe/transportes/caminos/normas_carreteras/documentos/manuales/Manual_de_Seguridad_Vial_2017.pdf?fbclid=IwAR37pfT4iC6gUom_o-0ceucgVt2qwlrX5ICH7YgvwSvEIEjAMLfjimyo1P8 (accesed on 30/03/21)

2. Gershon P, Ehsani J, Zhu C, O'Brien F, Klauer S, Dingus T, et al. Vehicle ownership and other predictors of teenagers risky driving behavior: Evidence from a naturalistic driving study. Accid Anal Prev. 2018;118: 96-101.

---

## [Decision Letter · Decision Letter 1]

4 May 2021

Use of seat belts among public transport drivers in Tacna, Peru: Prevalence and risk factors

PONE-D-20-30480R1

Dear Dr. Miñan-Tapia,

We’re pleased to inform you that your manuscript has been judged scientifically suitable for publication and will be formally accepted for publication once it meets all outstanding technical requirements.

Kind regards,

Feng Chen

Academic Editor

PLOS ONE

Additional Editor Comments (optional):

Reviewers' comments:

Reviewer's Responses to Questions

**Comments to the Author**

1. If the authors have adequately addressed your comments raised in a previous round of review and you feel that this manuscript is now acceptable for publication, you may indicate that here to bypass the “Comments to the Author” section, enter your conflict of interest statement in the “Confidential to Editor” section, and submit your "Accept" recommendation.

Reviewer #1: All comments have been addressed

Reviewer #2: All comments have been addressed

2. Is the manuscript technically sound, and do the data support the conclusions?

Reviewer #1: (No Response)

Reviewer #2: Yes

3. Has the statistical analysis been performed appropriately and rigorously? 

Reviewer #1: (No Response)

Reviewer #2: Yes

4. Have the authors made all data underlying the findings in their manuscript fully available?

Reviewer #1: (No Response)

Reviewer #2: Yes

5. Is the manuscript presented in an intelligible fashion and written in standard English?

Reviewer #1: (No Response)

Reviewer #2: Yes

6. Review Comments to the Author

Reviewer #1: (No Response)

Reviewer #2: (No Response)

7. PLOS authors have the option to publish the peer review history of their article (what does this mean?). If published, this will include your full peer review and any attached files.

Reviewer #1: No

Reviewer #2: No

---

## [Editor Report · Acceptance letter]

6 May 2021

PONE-D-20-30480R1 

Use of seat belts among public transport drivers in Tacna, Peru: Prevalence and risk factors 

Dear Dr. Miñan-Tapia:

I'm pleased to inform you that your manuscript has been deemed suitable for publication in PLOS ONE. Congratulations! Your manuscript is now with our production department. 

Kind regards, 

on behalf of

Dr. Feng Chen 

Academic Editor

PLOS ONE